# Tradition and Innovation in the Italian Wine Industry: The Best Practices of Casa Paladin

Daniele Grechi [1,*], Enrica Pavione [2], Patrizia Gazzola [2] and Francesca Cardini [2]

1 Dipartimento di Diritto, Economia e Culture, University of Insubria, Via. S. Abbondio, 12, 22100 Como, Italy
2 Dipartimento di Economia, University of Insubria, 21100 Varese, Italy; enrica.pavione@uninsubria.it (E.P.);
patrizia.gazzola@uninsubria.it (P.G.); fcardini@studenti.uninsubria.it or cardini.francy@gmail.com (F.C.)
* Correspondence: grechi.daniele@uninsubria.it

**Abstract:** This study aims to make a significant contribution to the development of a model for integrating research in the wine sector, innovative knowledge, and family businesses with the traditional mode of production in the context of the introduction of modern production technologies with a view to sustainability and the improvement of services in tourism. From a methodological point of view, the research is based on the case study and, in particular, on Casa Paladin, a family business in the Italian wine sector, which bases its strategy on innovation in production processes and customer relations to obtain high product quality with the aim of meeting consumer needs. The findings underscore the crucial role of family businesses in preserving cultural and traditional elements in the wine sector, with technology and innovation serving as vital drivers for their development. Casa Paladin's commitment to innovation in production processes and products is evident, emphasizing sustainability as a core element that impacts customer relationships and product quality. Enotourism, including tastings, festivals, and fairs, emerges as a significant aspect contributing to the promotion of the company's history, culture, and traditions. This study posits Casa Paladin as a notable example in the Italian wine industry, offering transferable insights for other businesses. Its successful integration of culture, innovation, and sustainability contributes to a broader understanding of the contemporary role of family businesses in the Italian wine sector.

**Keywords:** wine; wine industry; family business; case study





## 1. Introduction

The wine industry is one of the most representative markets globally and is primarily composed of small to medium-sized enterprises operating at the local, regional, or national levels. Wine production is one of the oldest and most expressive economic activities in many countries, featuring a variety of high-quality products that embody the passion, traditions, and craftsmanship of the producers [1]. Considering the actual data [2], based on the information of 29 different countries that represent 91% of the total production of wine for 2022, it is estimated between 257.5 and 262.3 Mio hl, despite in the European Union (EU), a series of climatic adversities (spring frosts, hailstorms, excessive heat, and drought) occurred throughout the entire vegetative period of 2022. Italy continues to rank as the world's leading producer, with an estimated volume of 50.3 million hectoliters, consistent with the wine production of 2021.

The data are represented in Figure 1.

In the Italian wine sector, it is usual to find family companies. Wine, in fact, does not qualify as a universal and anonymous consumer good; in fact, its consumption is influenced by a series of cultural factors such as values, symbols, and traditions entrenched at the local level [3].

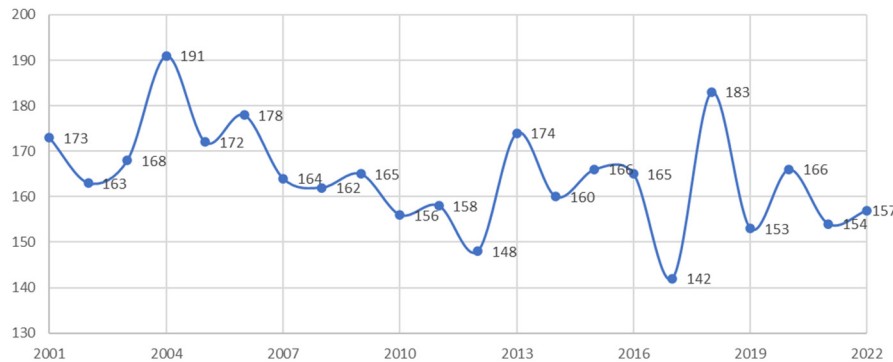

**Figure 1.** Wine production Eu27—Period 2001–2022 [2].

Family enterprises are characterized as businesses controlled and operated by a family, where crucial financial and operational decisions are shaped by its members, potentially in a sustainable manner across generations [4]. Unlike the traditional focus solely on financial value, the family-run business model encompasses both economic and non-economic dimensions of value acquisition. The pursuit of socio-emotional wealth stands out as a defining element for small family enterprises, driven by the desire to not only create and share their story but also establish profound connections with their final customers [5]. With this premise, this paper is structured as follows. In the first part, the phenomenon of wine-related activities is analyzed in relation to generations and family businesses, examining the values derived from this type of enterprise. Subsequently, the importance of technology and sustainability in wine production is explored, leading to the discussion of a case study linked to a company that embodies all these characteristics and elements. The case study was qualitatively examined through documents internal and external to the company, as well as through structured interviews. The first interview was conducted with Martina Paladin, export manager at the winery Casa Paladin. Successive interviews were conducted with other members of the family to obtain more information.

## 2. Methodology

In this study, a mixed-methods approach was adopted to conduct a comprehensive examination of the subject matter, with a particular focus on Casa Paladin as a case study [6].

A methodology based on a single case study is reliable [7,8] if the case study chosen is "extreme, unique, revealing, and pioneering". Casa Paladin can be considered as such, since the company is a successful firm that produces a wide range of high-end wines and presents an innovative approach to sustainability policies.

Although the data and information used in case studies may come from a variety of sources [9], the tool of conducting an interview [10] was chosen to write this paper.

A central component of our methodology involved the implementation of semi-structured interviews with key stakeholders, including representatives from Casa Paladin. A comprehensive set of questions was devised to explore the intricacies of Casa Paladin. In addition to interviews, our research drew from an array of sources, including public documents, the company's website, and insights gathered directly during the interview process. The interview was thoughtfully designed to extract in-depth perspectives on the company's marketing strategies, historical trajectory, and its intricate relationship with the fields of tourism and innovation. Open-ended questions were carefully crafted to facilitate candid and nuanced responses from the interviewee [11]. To further enrich our qualitative analysis, we conducted an examination of relevant documents and materials associated with Casa Paladin's marketing initiatives, historical background, and role in the tourism and innovation sectors. This document analysis, in conjunction with the insights gleaned from interviews, forms the basis of our case study.

In light of these considerations, the present study aims to examine the central role of family businesses in the wine sector. In particular, through a case-study approach, the

research explores how the interaction between culture, tradition, tourism, technology, and innovation influences the sustainability of business practices. The aim is to highlight good practices that can be transferred in general to family-run wineries.

## 3. The Role of Family Business in the Wine Industry

As defined by the authors of [5], "A wine business for this research is one whose core activities are the making, marketing and selling of wine and does not include a business that only grows grapes".

Starting from this definition, it is important to briefly define the role of family business from an overall point of view.

There are several authors that offer a valuable classification that sheds light on the distinct characteristics inherent in business models within the context of family enterprises [3,12–14]. This classification not only holds relevance for a broad spectrum of family businesses but also proves particularly insightful when applied to family-owned ventures in the wine industry.

The wine industry frequently features family businesses [15], where a distinct connection to a specific wine is formed in line with the family winery's values, symbols, and traditions [4]. As sustained by Parmentier [16], the concept of a family brand, defined as the collection of associations linked to a specific family, holds considerable sway over customer perceptions. This influence is particularly pronounced due to the competitive edge and market sway derived from the family's historical legacy and heritage within the family wine business [17,18]. Family wine enterprises harness resources such as the family name, family-owned real estate, and family heritage to craft their identity [19,20]. As a result, family wineries infuse their products with symbolic attributes, impacting sales growth through the symbolic value associated with the family name, a factor especially pertinent in the wine industry [13]. Given that the winemaker is often a family member, the family name serves as a critical branding dimension at both the corporate and product brand levels [21].

Once the corporate reality is established, it should be important to have a view that will be able to ensure the future of the business. Succession planning involves the transfer of management and control of corporate ownership through generations [22]. In this field, family-owned wineries face intergenerational challenges that require significant preparation to be addressed appropriately and with the most suitable techniques [3]. Continuous adjustments and renewals within the company are essential in light of changes brought about by an increasing number of stakeholders and the introduction of innovations in the methods of production and sale of the goods and services related to their main activities [23].

One of the most important aspects of preserving the company's tradition is the sharing of the corporate vision in combination with internal knowledge between different generations [24]. This aspect allows the business to improve its efficiency and, consequently, its productivity, leading to a reduction in training costs. Beyond fostering increased trust and loyalty from customers, this approach plays a crucial role in shaping a distinctive corporate brand image. By adopting this strategy, businesses can effectively differentiate themselves from competitors, aligning with the insights of Craig, Dribell, and Davis [25]. Essentially, narrating the enterprise's journey and spotlighting its protagonists become potent tools for cultivating a positive and unique identity within the competitive landscape.

Family entrepreneurial activity, based on readily available knowledge, can support and encourage business continuity in rapidly changing environments. Nevertheless, it is advisable for succeeding generations to seek evolution through exploring new directions and decisions aimed at technological innovation and sustainability, an increasingly relevant theme in today's context [26].

Initiating this process necessitates a strategic, multi-year approach, whereby entrepreneurs meticulously plan the timing and appropriate modalities for enabling prospective successors to acquire fundamental functions and responsibilities pertinent to the

business [27]. This entails the identification and grooming of future leaders, the gradual delegation of escalating responsibilities to younger family members, and the phased dissemination of knowledge and competencies. Another relevant aspect underscores the significance of successor training and the assimilation of tacit knowledge. In this scheme, constructive dialogue assumes a pivotal role, fostering collaborative dynamics among family members and nurturing motivation and zeal towards achieving a shared objective [28]. Ultimately, due consideration must be given to the welfare and interests of all family members, reframing the generational transition not merely as a transference of power and control but as an opportunity to establish a wholesome and sustainable work environment for successive generations [29]. In family businesses, the concepts of rarity and inimitability play a significant role. Competitive advantage is thus justified by the implementation of rare and valuable resources, especially those that are difficult to imitate and duplicate, stemming from the family organizational culture [30]. Furthermore, a brief excursion in the context of wine tourism is mandatory. Family businesses play a pivotal role in shaping the overall experience [31]. The close-knit nature of family enterprises adds a personal touch to the interactions between tourists and winemakers. Visitors often appreciate the warm and familial atmosphere created by these businesses, which enhances the sense of connection to the local culture and traditions [32,33].

Family-owned wineries, with their generational expertise passed down through the years, contribute significantly to the authenticity of the wine tourism experience. The commitment and passion of family members involved in the winemaking process become integral components of the corporate image. This familial involvement fosters a sense of trust and reliability among consumers, as they perceive the dedication to quality and tradition as part of the family legacy [34].

Moreover, family businesses often embody a commitment to sustainable practices and a deep respect for the land. This emphasis on environmental stewardship adds another layer to the positive image of these enterprises, aligning with the increasing consumer interest in eco-friendly and socially responsible business practices [35].

For these reasons, wine tourism represents an opportunity to enhance wine regions, promote wine culture, support the local economy and products, and offer tourists an authentic and engaging experience in close contact with people, companies (family companies), and the territory.

## 4. Sustainability and Innovation in the Wine Industry: Challenges, Opportunities, and Communication Strategies

Recently, in the eno-gastronomic field, a cultural revolution has unfolded concerning the modern perception of food and beverages [36]. Beyond their nutritional function, they now embody ethical and sociocultural values. Consumers prioritize quality, origin, health, and safety, taking into account human rights and the environment [37]. Sustainability is paramount, involving environmentally friendly practices, equitable approaches, and competitive and economical production.

In the wine case, innovation strategies have emerged as one of the most crucial aspects of the wine industry, particularly for family-owned businesses. These strategies provide a means for companies to gain competitive advantages, foster long-term growth, and ensure the success and survival of the enterprise across generations. The impacts of innovation encompass cost reduction, increased productivity levels, enhanced product quality, and the development of new logistics, packaging, and storage methods [38,39]. Through innovation, companies can transform novel ideas into advantageous opportunities, leading to the introduction of new products into the market and an increase in overall business while adapting to evolving consumer needs. Customer perception may also involve variables such as price and packaging, both extensively considered in the product innovation process [40]. Italian wineries primarily innovate by introducing a diverse range of traditional products and maintaining limited production volumes to improve wine quality. The central focus is on the perception of the company's image, while process

innovations are internally managed with flexibility [41]. Noteworthy are technological innovations and restructuring efforts that preserve the historical heritage of cellars, directly linked to recreational or tourist phenomena [42]. Finally, distribution and promotion are often handled directly by the family, and in the case of exports, distributors are considered "wine ambassadors". The use of advanced technologies contributes to the creation of unique products, ensuring a competitive advantage through superior quality, efficiency, and customer satisfaction [3].

Therefore, it is evident that innovation in the wine industry is a central element for corporate survival and addressing emerging competitiveness. The family business must continually activate new emotional resources that enhance the recognizability of the corporate brand and enable a significant competitive advantage through product differentiation strategies. The physical environment of wineries, as well as the virtual one on websites and social networks, must have a positive effect on customer perception and create an atmosphere of trust and satisfaction, contributing to the building of a long-term relationship with the consumer [43].

The innovative process of winemaking is not always smooth, hindered by gaps in information and technical knowledge that can delay or postpone the adoption of sustainable practices. Bridging knowledge gaps through continuous education is crucial to accelerating the implementation of innovative solutions [44]. Organic practices, while promoting sustainability, often incur higher production costs, especially with certification pursuits, and consumers may not readily pay the price difference. Producers must effectively communicate the value of these changes, emphasizing the quality and properties of the product. Balancing the challenges of innovation, knowledge dissemination, and cost considerations is essential for a successful transition to sustainable and innovative winemaking practices [45].

Among the countless solutions and innovative techniques related to sustainability, we find the promotion of natural solutions that allow for the biological control of the spread of parasites and bacteria in vineyards. This involves abolishing the use of chemical pesticides or synthetic fertilizers in favor of natural and eco-friendly products. This is followed by the implementation of ecological engineering works focused on pollination areas and the filtering of effluents from vineyards. These efforts are accompanied by the management of water resources, enabling efficient water use through controlled irrigation systems and rainwater harvesting [46].

New trends are also evident in the field of transportation, where environmental considerations impact the reorganization and distribution of wines. Practices such as sustainable transportation of grapes and bottles help reduce $CO_2$ emissions, optimize energy efficiency in cellars, and utilize packaging made from recyclable and even biodegradable materials [47].

Another important aspect of disseminating values and information related to sustainability involves the support provided by social media and company websites. Through these platforms, companies can share their ideas regarding responsible practices and behaviors that individuals can adopt for their own well-being and that of the environment. Communication and engagement are crucial aspects of actively encouraging consumer involvement in promoting sustainable solutions.

In conclusion, we can assert that a sustainable approach is a significant driver for innovation in wine production and represents a valid response to the needs and expectations of consumers. Considering wine as a common good, the implementation of a sustainable strategy allows the family business to preserve its value by marketing quality products that help maintain the integrity of the territory. A sustainable approach, centered on values and awareness, enhances cultural and entrepreneurial knowledge and supports the business in identifying new solutions and business opportunities, ensuring long-term benefits and success across generations [47].

## 5. Case Study: Casa Paladin

In this paragraph, we have deeply analyzed the Casa Paladin company.

During this research, some public information, based on the corporate website and external material, was used in combination with an interview with the Marketing Manager of "Casa Paladin". This interview proved to be a valuable source of information, offering key insights into the company's background, history, and its dynamic relationship with the realms of tourism and innovation. The Marketing Manager shared valuable perspectives on Casa Paladin's marketing strategies, shedding light on the company's approach to embracing innovation in the context of the ever-evolving tourism landscape. This firsthand account from a key figure within the organization enriches our understanding of Casa Paladin's positioning and underscores the interplay between marketing, history, and the pursuit of innovative practices in the business. Casa Paladin is a brand that was established in 2017 and brings together five companies owned by the Paladin family. Observing the design of the company's logo, it is possible to see the presence of four wine bottles, which represent the four estates belonging to Casa Paladin, namely, Paladin Vigne e Vini, Bosco del Merlo, Premiata Fattoria di Castelvecchi, and Castello Bonomi. Furthermore, Casa Paladin has established a collaboration with a fifth company, called Casa Lupo.

In Figure 2, it is possible to see the entire range of brands that are in charge of Casa Paladin.

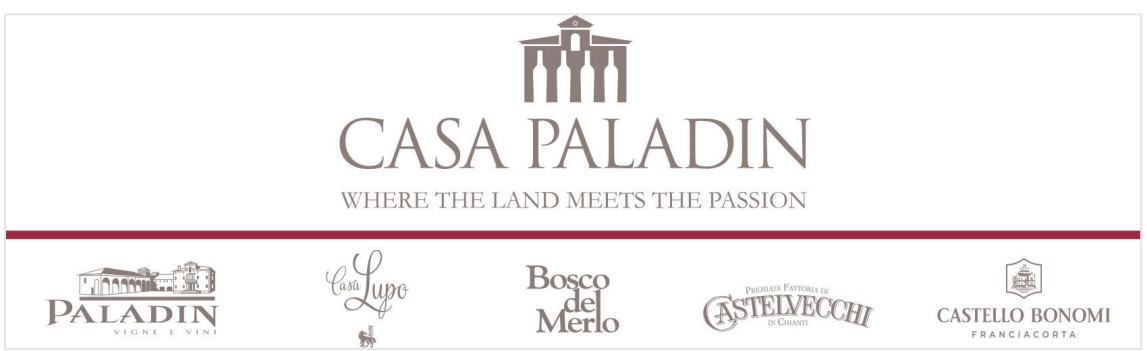

**Figure 2.** Brands in Casa Paladin. Source: website Casa Paladin, 2017 [48].

Starting with the historical point of view, Casa Paladin was established in 1962 in Motta di Livenza near Treviso and was later relocated to Annone Veneto, where, actually, the main headquarters is now located. The company was founded by Grandfather Valentino Paladin, who began planting the first vines and producing bulk wine in demijohns. Then, in the 1970s, he focused on bottling and creating the first labels, eventually giving rise to what is now an all-encompassing business company [48]. Subsequently, in 1977, a new company was created (Bosco del Merlo) that was defined as "a Veneto company with a Friulian identity" [49]. The name is derived from local topographic maps and recalls the oak forests that once covered the area. The core value to which the company is committed is the research and implementation of sustainable practices to best express the identity and potential of the territory, with all its nuances. The brand is dedicated to achieving excellent winemaking quality, allowing the family to pass down and express its values and passions. In 2004, Casa Paladin decided to acquire another business from Tuscany (Premiata fattoria di Castelvecchi). The winery boasts the record of being the oldest still productive in the entire Chianti region, and it was built in the dungeons of Castelvecchi Castle around 1043 A.D. [50]. Moreover, in 2008, there was the acquisition of a company named Castello Bonomi in Franciacorta, a region where viticulture has shaped human evolution and influenced habits over the centuries [51]. The Castello Bonomi estates are located on the slopes of Mount Orfano, in a spectacular natural amphitheater in the municipality of Coccaglio (northeast of Lombardy), and their vineyards are divided into 39 plots, including 6 different crus. Grapes are harvested and processed separately to enhance the unique characteristics and identities of each terrain through the subdivision into microzones.

Starting in 2016, Casa Paladin has collaborated with Casa Lupo, a Venetian company in Valpolicella, a region with a high viticultural vocation that has been producing wines symbolizing Italian excellence internationally for years [52]. By focusing on management and its implications, the relevance of the family business becomes clear. In fact, the definition of the roles assigned to each family member is based on each individual's predisposition. The Paladin family has planned a division of roles based on both the different stages of production and the multiple geographical areas involved in the business. Casa Paladin is involved and exports all over the world in the principal oenological markets (with a focus on North America and Europe). The philosophy of Casa Paladin is centered around sustainability [53]. They actively research and develop sustainable production, involving winemakers, universities, and agronomists. The mission embraces both the productive and social aspects. Certifications such as "International Food Standard" and "VEGAN OK" attest to compliance and the absence of animal-derived products. In 2021, Paladin and Bosco del Merlo obtained EQUALITAS certification for Sustainable Wine, that is related to social, environmental, and economic aspects. The ethics of sustainability are reflected in every stage of production, from harvesting to bottling. Casa Paladin's commitment to sustainability preserves biodiversity, improves soil quality, and promotes responsible tourism, ensuring product quality. This commitment is based on preserving past traditions, enhancing current practices, and safeguarding future viticulture.

The philosophy of Casa Paladin is based on four pillars that are represented in Figure 3.

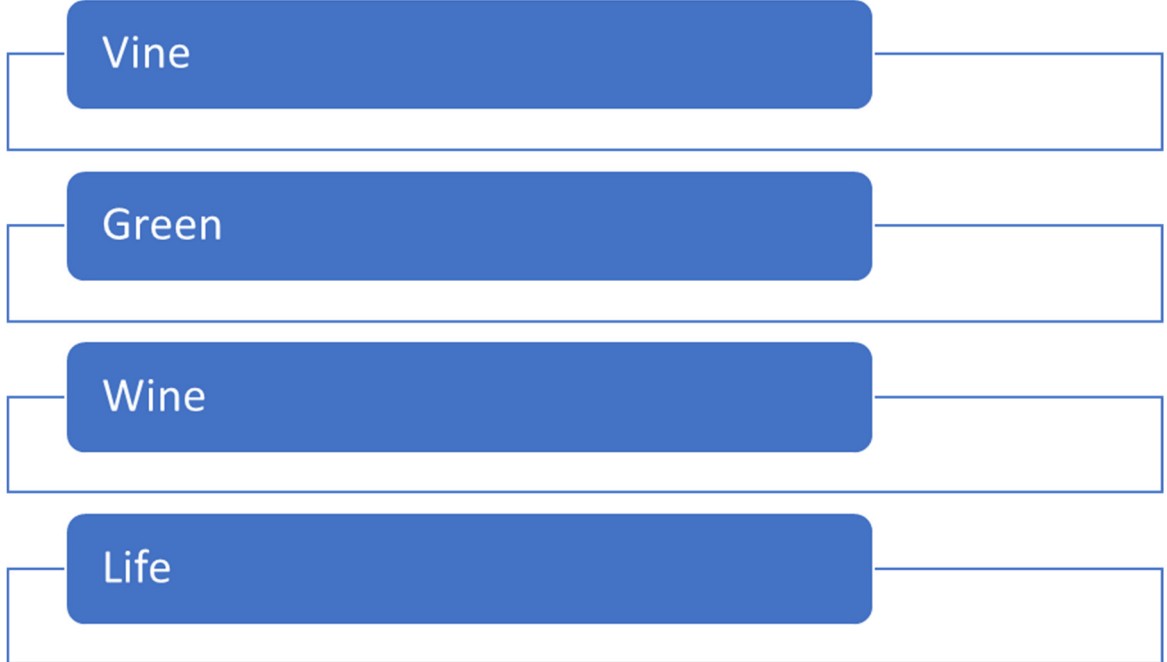

**Figure 3.** Pillars of Casa Paladin.

The four V project (in Italian, "Progetto 4V": Vite, Verde, Vino, and Vita) summarizes the main missions of the company, which consist of respect for the Vine, the safeguarding of the Green, the production of Sustainable Wine, and finally, the protection of Life. The company's culture is also based on four fundamental pillars.

First, Casa Paladin is committed to preserving the uniqueness of its origins. This entails safeguarding the heritage of its family, land, and cultural heritage. Through the conservation of ancient mother plants, the company not only honors its past but also revitalizes vineyard landscapes, rediscovering the authenticity and distinctiveness of its wines. Second, there is a strong emphasis on respecting the character of the territories and the natural integrity surrounding the vineyards. By prioritizing sustainable practices and biodiversity conservation, Casa Paladin ensures the longevity of its ecosystem while

nurturing a harmonious relationship between nature and viticulture. Additionally, Casa Paladin places great importance on fostering a sense of community and collaboration. Through initiatives such as educational programs and partnerships with local organizations, the company actively engages with its surrounding communities, contributing to social cohesion and economic development. Furthermore, innovation is ingrained in Casa Paladin's culture, driving continuous improvement and adaptation to evolving market trends and consumer preferences.

Production rhythms and harvest times do not chase the frenetic pace of the market but rather respect the cycle of nature and the characteristics of the territories [54]. The company actively supports social initiatives, including the "Life in Rosè" project aimed at raising awareness and combating breast cancer. Through collaborations with Bosco del Merlo, the company contributes to the Pink Ribbon Campaign by LILT, focusing on breast cancer prevention and patient support. The 'Life in Rosè' initiative includes sponsorship of the Treviso in Rosa event, with proceeds from sales of specific wines dedicated to breast cancer research [55]. Casa Paladin's dedication to culture is evident in partnerships with renowned artists, exemplified by the Vineargenti Rosso Riserva project with Fabrizio Plessi [56] (Website Casa Paladin—Style). These collaborations reflect the company's commitment to preserving family traditions, promoting art, and producing authentic, quality products while respecting the environment and territories.

## 6. The Role of Wine Tourism and Events for Casa Paladin (Interview with Martina Paladin)

Wine tourism is a fundamental element for Casa Paladin and one of the most important channels for the company's growth. For consumer satisfaction, wine must create an authentic experience and narrate the story and traditions of the family [57]. As previously explained, in a broad view, in the theoretical part [58], the phenomenon of wine tourism involves both the territories of Veneto, including the areas of Brescia and Franciacorta, and the lands of Tuscany. Different target audiences are found in various regions; for example, in Tuscany, the audience is primarily international and consists of consumers who appreciate Chianti and Made in Italy. The scenic settings and internationally recognized typical products are valued for their quality, craftsmanship, and design. Every year, Casa Paladin organizes a series of events at its estates to enhance the tasting experience. Among the numerous activities offered, there is "Cinema in Cantina" [59], an experience that allows families to watch a movie or documentary while exploring the flavors of the presented wines. Another opportunity for a unique experience is "Yoga in Cantina" [60], a meditation session amidst the vineyards in Veneto and Tuscany, followed by an aperitif and tasting of local products. Picnics among the vines and on the panoramic terraces of the cellars, where picturesque sunsets unfold amid the woods and olive groves, are also part of the offerings. There are treasure hunts, concerts, gourmet events, vineyard walks, and many other initiatives that guide consumers in discovering the four estates. By engaging in wine tourism, Casa Paladin supports its family culture, enhances the wine-growing territory, and allows tourists to visit and appreciate its cellars and products through guided tours and tastings. The activities offered provide opportunities for unique sensory experiences and foster the sharing of knowledge, stories, and traditions among producers, experts, and wine enthusiasts from around the world.

## 7. Marketing, Innovation, and the Role of Its Website for Casa Paladin (Interview with Martina Paladin)

In the pursuit of enhancing process and product quality, Casa Paladin consistently directs substantial investments toward technological and innovative endeavors. Notably, amid the COVID-19 pandemic, the company implemented pivotal adjustments that bolstered work performance and heightened productivity, and among the many innovative solutions was the establishment of a novel warehouse type equipped with an automated door opening and closing system. This strategic modification mitigates the risk of prolonged door openings during goods unloading and product shipping, thereby averting

a potential reduction in internal temperature and safeguarding the integrity of the wine preservation process. Furthermore, the warehouse capacity has been expanded twofold, accommodating a significantly increased volume of bottle storage. Concurrently, a photovoltaic system installation has been enacted to optimize the reuse of generated energy, resulting in heightened efficiency and a consequential reduction in costs and consumption. Finally, a pioneering bottling methodology has been developed, doubling the production rate from 3000 to 6000 bottles per hour. This centralized and automated bottling process incorporates sensors and photocells adept at identifying errors or irregularities, such as the presence of unlabeled bottles or insufficient filling levels, thereby ensuring the quality and integrity of the final product. These operations were decided by the company to minimize any form of human error, which could lead to significant financial costs and inaccuracies in resource management. Casa Paladin's objective is to optimize processing efficiency by increasing production speed, enhancing product quality, reducing waste, and promoting sustainability. Despite Casa Paladin being a historic and internationally recognized company, marketing and websites play a crucial role in ensuring that the company is known to an increasingly larger number of customers and wine enthusiasts. Within the company, there is a structured marketing office where one person is dedicated to enotourism management and a second individual oversees marketing management. Casa Paladin manages various digital platforms, each dedicated to one of the five companies that constitute the brand. On each website, one can find information about the history of the different brands, their philosophy, and the main activities carried out at the estate's cellars. This way, for both loyal customers and occasional tourists, it is possible to stay constantly updated and view the company's offerings regarding traditional products and novelties. The social media aspect, distributed across Instagram and Facebook channels, is managed by an external company. Social media presence is essential for customer attraction and requires a captivating and structured strategy to reach the final clientele, segmented based on geographical areas and typical customs, with the aim of maximizing the effectiveness of communications and creating an emotional bond with the target audience. Through social media, the company shares its story and traditions, communicates upcoming events, and introduces new developments. A central figure for communication and the success of the company is the enoblogger [61], a person in direct contact with the end consumer, making wine accessible to everyone through small, intriguing posts. The enoblogger's role is to share information, experiences, and oenological advice to educate and inspire followers, leading them to discover new wines, regions, and tasting techniques. In relation to the development of new trends, the promotion of the food-pairing phenomenon becomes particularly relevant. Casa Paladin proposes pairings between wine and foods such as fish dishes, charcuterie boards, or seasonal fruits. Additionally, events and parties are organized to offer consumers the opportunity to discover new pairings and fully appreciate the flavors of food and beverages. There is also a segment dedicated to newsletters, a form of digital communication through which regular updates, advice, and news are sent to a group of customers who have voluntarily subscribed to the distribution list. This interaction method allows for the receipt of discount codes and promotions for events, such as birthdays, and recurring celebrations. The option of having a loyalty card to accumulate points, ensuring additional benefits and rewards for the consumer after a certain number of purchases, is also available. For Casa Paladin, the use of social media and digital communication channels is essential for brand image creation and constant, direct interaction with the public. Through these tools, the company promotes its products and services, facilitates online sales, and attracts an ever-growing number of consumers.

## 8. Conclusions, Limitations, and Further Developments

The intertwining of innovation and tradition in the viticultural realm is integral, and this study, anchored in an Italian enterprise, aims to delve deeper into how the familial business model can serve as a pivotal tool in advancing this industry in innovative and productive ways. In this context, the analyzed case study confirms the role of implicit

values in the family business (such as tradition, history, and corporate philosophy) that have constituted the driving force towards a sustainability-oriented strategic approach, which has become an integral part of the business model. The company's ability to successfully integrate culture, innovation, and sustainability provides original insights that can be transferred to other business contexts, contributing to a broader understanding of the role of family businesses in the contemporary Italian wine industry. It will be interesting to examine and evaluate the company's results from a medium-term perspective to fully appreciate the strategic choices made.

Overall, the company's success depends on a delicate balance between tradition and innovation. Ignoring tradition in favor of innovation can potentially lead to the loss of values that consumers associate with a brand. On the other hand, favoring tradition over innovation can mean preserving cherished values but losing the benefits of innovation. Navigating a challenging economic, political, and social landscape requires companies to adapt quickly, and the synergy of tradition and innovation is of utmost importance to maintaining a competitive advantage. However, it is worth noting that the company's informal approach to innovation process management has limitations. While this informal structure is effective now, it can create barriers to future growth and prevent the integration of external managers into the innovation function.

Although the company has not provided economic and financial data that can concretely testify to the effect of the strategic choices made, the interviews carried out highlight a series of positive and synergistic effects linked to the combination of tradition and innovation for the benefit of sustainability. The positive results, in terms of improvement of soil quality, energy savings, enhancement and connection with the local community, and sharing of knowledge and traditions, represent the declination of good practices that can also be replicated in other realities.

In addition, innovation and, in particular, investments made in technology have allowed the company to enhance work performance and increase production while reducing waste and ensuring product quality.

The result is a virtuous circle, where the identity and value elements of the family business feed a circuit of innovative and sustainable initiatives, allowing the company to preserve the value and integrity of the territory while ensuring the production and marketing of high-quality products. This type of approach improves cultural and entrepreneurial knowledge and fosters new business opportunities in the long term, as well as through future generations. Additionally, the company places significant emphasis on catering to tourists, employing dedicated personnel for online marketing and organizing guided tours with skilled professionals in their territories. Wine tourism plays a pivotal role in their overarching strategy, serving as a cornerstone for attracting visitors and fostering engagement with their brand. There are also some other key points. For example, climate change poses a significant threat with its potential to disrupt vineyard ecosystems and compromise wine quality through extreme weather events and shifts in temperature patterns. Moreover, market competition presents challenges as producers try to differentiate their products among consumer demand for sustainably produced wines.

The importance of this study lies in its implications, which highlight the central role of innovation throughout the production and supply chain. The purpose of this study is to provide valuable insights into innovation management as a resource for owners and managers of family wine businesses who seek to improve their business. Therefore, we believe that this study, despite facing the limitations of a case study, may have significant implications for the literature on sustainability in family businesses, identifying their peculiarities and key results.

A single case study has its limitations; for this reason, possible future research could benefit from a comparative analysis that includes other family businesses in the wine industry to enrich and confirm our findings. For this reason, a structured set of interviews could be the basis for the creation of a dataset that contains a relevant number of best practices, innovation ideas, and strategy mixes concerning this field.

**Author Contributions:** Conceptualization, E.P. and F.C.; methodology, D.G. and P.G.; validation, D.G., P.G. and E.P.; formal analysis, E.P., D.G. and F.C.; investigation, F.C. and E.P.; resources, E.P. and F.C.; writing—original draft preparation, E.P., D.G., P.G. and F.C.; writing—review and editing, D.G., P.G. and E.P.; supervision, E.P. and P.G.; funding acquisition, D.G. and P.G. All authors have read and agreed to the published version of the manuscript.

**Funding:** The APC was funded by the University of Insubria. This paper (D.G.; P.G.) is part of the project NODES, which received funding from the MUR—M4C2 1.5 of PNRR with grant agreement no. ECS00000036.

**Institutional Review Board Statement:** Not applicable.

**Informed Consent Statement:** Written informed consent has been obtained from the interview to publish this paper.

**Data Availability Statement:** The data presented in this study are available on request from the corresponding author.

**Acknowledgments:** We acknowledge Casa Paladin for its support and availability.

**Conflicts of Interest:** The authors declare no conflicts of interest.

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
