# Peer review of "Tradition and Innovation in the Italian Wine Industry: The Best Practices of Casa Paladin"

_sustainability, doi:10.3390/su16072857_

Round 1
Reviewer 1 Report
Comments and Suggestions for Authors
The paper is about family business and most importantly the tradition of Paladian Winery. The paper despite its flows has merits because it focuses on family business dynamics and the exploration of balancing tradition and innovation.
The flows stand on the methodology, which is not clearly described. In thick lines, the methodology adopted in this study is a mixed-methods approach, based on semi-structured interviews with key stakeholders, including representatives from Casa Paladin. Furthermore, document analysis was performed delving into documents and materials associated with Casa Paladin's marketing initiatives, historical background, and its role in the tourism and innovation sectors complements the interview data, providing a more comprehensive view of the case study. What I would ask as a reviewer is more details on the methodology and also a more detailed description of
Author Response
Dear Reviewer,
in the file there are our answers.
best
The Authors

Reviewer 2 Report
Comments and Suggestions for Authors
Title: „Tradition and innovation in the wine industry: the best practices of Casa Paladin“
- This paper correspond for scope of journal.
- The title corresponds to the content of the paper.
- Abstract is no structured. Should be improve contents of abstract with clear expression of aim of study, method of study, main results which were obtained in research, and conclusion sentence.
- This study represents a significant contribution to developing model of integration of research in wine industry innovative knowledge and family business with the traditional way of production, in frame of introduction of modern technology of production in function of sustainability and improving services in tourism.
- The main question of paper addressed to identifying role and potential of family business in Italian wine sector. This study conducted on of Casa Paladin practice which based on innovation in production processes and relationships with customers to achieve high quality of product in the aim to satisfy requirements of consumers.
In study is indiced that Enotourism, including tastings, festivals, and fairs, represents as a significant event and program of activities contributing to the promotion of the company's history, culture, and traditions
- The aim of research is not clearly and fully pointed in chapter of Introduction. In structure of scientific paper is rule to write aim of study on the end of chapter of Introduction. It should be particular (last) paragraph of the chapter of Introduction .
- Key words are appropriate.
- The aim of research is not clearly and fully pointed at the end of chapter of Introduction. The aim should be last paragraph of chapter of Introduction!
- The text in the last paragraph of the Introduction from lines 56 to 63 is not appropriate for the Introduction chapter. Some sentences can be used in the chapter Material and methods of research. So authors can use them for the Material and Methods chapter.
- In the chapter Material and methods from lines 233 to 236 is pointed out aim of study but aim of study is not appropriate for the chapter Material and methods. The objective should be moved to the Introduction chapter.
- Scientific methodology is presented partly correct for this type of study.
- The specific results achieved by Casa Paladin are not shown. It would be necessary to show the comparative results of changes and progress of Casa Paladin in the development period of the company and depending on the application of innovative business methods. Also, comparison of results with other companies.
- The conclusions are not completely written on the base of presented results!!! The rule is that in conclusion is not allowed quotation of attitudes, opinions, assertions other authors, because it is not results of study in this research article.
- The text in the first paragraph of theConclusion from lines 414 to 417 and in the second paragraph from line 418 to 421 is not appropriate for the chapter of Conclusion. It should be delete.
- Manuscript is acceptable after minor corrections!
Author Response
Dear Reviewer 2, Thank you for your comments.
The changes are in the file.
best
The authors

Reviewer 3 Report
Comments and Suggestions for Authors
Dear Authors,
The article entitled "Tradition and innovation in the wine industry: best practices from Casa Paladin" aims to explore how the interplay of culture, tradition, tourism, technology and innovation affects the sustainability of business practices.
I consider the choice of the research topic to be very relevant to the development of wine tourism. The research results and recommendations presented can help improve the operation of wine businesses in less developed regions.
After reading the article, I have the following comments and suggestions for its improvement:
Abstrakt
I suggest improving the abstract according to the requirements of the journal Sustainability, reduce unnecessary descriptions, introduce the research methods used, but expand the results section.
Titel
The titel of the paper should be modified. It lacks reference to the research area (country).
Introduction
The introduction to the topic was not based on international literature. The questions were not answered: why did the authors take up this study? What benefits can flow from this study? And what is the status of this research? The article does not pose specific research questions. At the end of the chapter, the purpose of the work should be outlined. The introduction should be numbered 1.
Methodology
This chapter should be thoroughly revised.
The research stages should be described in detail.
I suggest introducing:
- research area-location map, characteristics,
- What the survey looked like, what questions it contained?
- Who was the survey questionnaire?
- What source materials were used in the research process?
Results
There is a lack of detailed analysis and interpretation of the research results.
In order for the article to meet the standards of a scientific paper, the authors should compare the results of the study with other results of other scientific papers and set recommendations. Especially since the authors refer to sustainability in the theoretical part. Meanwhile, there are no references in this direction. I suggest completing the following:
- what elements of the wine business are most important for sustainable development?
- what are the barriers and threats to development?
- what are the strengths of development?
[302] Casa Paladin's philosophy is based on the four pillars shown in Figure 3.
I suggest a more detailed description of this figure.
Discussion
This chapter still needs to answer the question: what tangible benefits has this study brought to the development of tourism in rural areas and make recommendations. Highlight which elements of cultural heritage are most important.
Correct literature according to journal rules
Kind regards,
Author Response
Dear Reviewer 3,
our changes are in the file.
best
The authors

Round 2
Reviewer 3 Report
Comments and Suggestions for Authors
The article was translated according to the reviewer's suggestions.